# Electrochemical Biosensors by Means of Molecularly Imprinted Polymers (MIPs) Cortisol Recognition

**DOI:** 10.3390/polym17040545

**Published:** 2025-02-19

**Authors:** Jindapa Nampeng, Naphatsawan Vongmanee, Chuchart Pintavirooj, Wen-Tai Chiu, Sarinporn Visitsattapongse

**Affiliations:** 1Department of Biomedical Engineering, School of Engineering, King Mongkut’s Institute of Technology Ladkrabang, Bangkok 10520, Thailand; jindapa.na@kmitl.ac.th (J.N.); naphatsawan.v@hotmail.com (N.V.); chuchart.pi@kmitl.ac.th (C.P.); 2Department of Biomedical Engineering, National Cheng Kung University, Tainan City 701401, Taiwan; wtchiu@mail.ncku.edu.tw

**Keywords:** molecularly imprinted polymer, biosensors, cortisol, electrochemical sensing, cyclic voltammetry

## Abstract

Depression and anxiety are two common mental health issues that require serious attention, as they have significant impacts on human well-being, with both being emotionally and physically reflected in the increasing number of suicide cases globally. The World Health Organization (WHO) estimated that about 322 million people around the world experienced mental illnesses in 2017, and this number continues to increase. Cortisol is a major stress-controlled hormone that is regulated by the hypothalamic–pituitary–adrenal (HPA) axis. The HPA axis has three main components, including the hypothalamus, pituitary gland, and adrenal gland, where cortisol, the primary stress hormone, is released. It plays crucial roles in responding to stress, energy balance, and the immune system. The cortisol level in the bloodstream usually increases when stress develops. Molecularly imprinted polymers (MIPs) have been highlighted in terms of creating artificial bioreceptors by mimicking the shape of detected biomolecules, making natural bioreceptor molecules no longer required. MIPs can overcome the limitations of chemicals and physical properties reducing over time and the short-time shelf life of natural bioreceptors. MIPs’ benefits are reflected in their ease of use, high sensitivity, high specificity, reusability, durability, and the lack of requirement for complicated sample preparation before use. Moreover, MIPs incur low costs in manufacturing, giving them a favorable budget for the market with simple utilization. MIPs can be formulated by only three key steps, including formation, the polymerization of functional monomers, and the creation of three-dimensional cavities mimicking the shape and size of targeting molecules. MIPs have a high potential as biosensors, especially working as bioanalytics for protein, anti-body, antigen, or bacteria detection. Herein, this research proposes an MIP-based cortisol biosensor in which cortisol is imprinted on methyl methacrylate (MMA) and methacrylic acid (MAA) produced by UV polymerization. This MIP-based biosensor may be an alternative method with which to detect and monitor the levels of hormones in biological samples such as serum, saliva, or urine due to its rapid detection ability, which would be of benefit for diagnosing depression and anxiety and prescribing treatment. In this study, quantitative detection was performed using an electrochemical technique to measure the changes in electrical signals in different concentrations of a cortisol solution ranging from 0.1 to 1000 pg/mL. The MIP-based biosensor, as derived by calculation, achieved its best detection limit of 1.035 pg/mL with a gold electrode. Tests were also performed on molecules with a similar molecular structure, including Medroxyprogesterone acetate and drospirenone, to ensure the sensitivity and accuracy of the sensors, demonstrating a low sensitivity and low linear response.

## 1. Introduction

Mental illness has become a global problem. The World Health Organization estimates that over 300 million people suffer from depression [1]. This especially became a problem during the COVID-19 pandemic, as COVID-19 caused physical illness and mental problems that affected people’s moods and behaviors. Stress and anxiety are generally experienced when people cannot fulfill their expectations, and they can produce significant changes in physical and mental health. A previous study stated that stress can cause changes in brain structures, cognitive function, and emotional function [2]. The hypothalamic–pituitary–adrenal (HPA) axis is responsible for hormone regulation and response to stressors [3]. The release of corticotropin-releasing hormones (CRHs) stimulates the anterior pituitary to release adrenocorticotropic hormone (ACTH), which, in turn, stimulates the adrenal cortex to produce glucocorticoids. Cortisol is a glucocorticoid hormone involved in blood glucose regulation, immunosuppressive effects [4], anti-inflammatory effects [4], and effects on the central nervous system [5].

Cortisol is a stress biomarker, as it plays a vital role in the stress-responsive neuroendocrine system [6]. Cay et al. [7] reported that the cortisol levels in saliva samples from volunteers increased significantly during stressful events compared to normal levels. Cortisol in the human body exists in the following two forms: protein-bound cortisol (cortisol-binding globulin, CBG) (90–95%) and free cortisol in plasma (5–10%) [6], and it can be found in sweat [7], saliva [8], serum [9], urine [10], and hair [11]. Traditional methods to detect cortisol levels include enzyme-linked immunosorbent assay (ELISA), colorimetric assay [12], lateral flow technology, and mass spectrometry [13]. However, these methods have limitations regarding their use in medical applications in terms of time, cost, and the pre-treatment process. A more efficient way to detect cortisol levels is to use a biosensor. A biosensor is a scientific device developed to detect and analyze chemicals or biological molecules by converting biological signals into measurable signals directly proportional to the concentration of molecules in the sample or inside the body. Biosensors have been developed and widely used in many research fields due to their specificity and selectivity for target molecules. Nowadays, biosensors are an essential tool for detecting and analyzing many bio-components or chemicals inside and outside the body. Blood glucose monitors for diabetes patients and urine test strips are examples of biosensors.

Molecular imprinting technology (MIT) is a molecular recognition technique that uses polymer synthesis to produce an artificial receptor compatible with a specific molecule. Molecular imprinted polymers (MIPs) are designed to mimic biological recognition, such as an enzyme and substrate or an antigen and antibody. This method has been widely used in various applications, such as pharmaceuticals [14], the food industry [15], drug delivery systems [16], health applications [17], nosocomial detection [18], and biosensors, due to its accuracy, low cost, and rapid detection capability. In the fabrication of an MIP-based biosensor, two functional monomers are combined with a cross-linker, initiator, and solvent to generate a polymer solution. The template polymer solution mixed with conductive particles is then polymerized under ultraviolet light. The template is later removed, leaving specific cavities on the surface. The MIP technique can provide a high sensitivity and selectivity for pseudoephedrine in biological samples [19] and produce a selective and accurate electrochemical chiral sensor for the sensitive and selective detection of other compounds [20]. MIPs can be used as biosensors to detect influenza A virus [21], estimate the agglutination tests of influenza subtypes [22], monitor SAR-CoV-2 in wastewater systems without biochemical techniques [23], recognize the cerebral dopamine neurotrophic factor (CDNF) protein [24], and detect the human epidermal growth factor receptor 2 (Her-2) protein, which is a significant breast cancer biomarker [25].

A cortisol biosensor based on molecular imprinting technology is the most promising line of active research for the detection of cortisol levels and has captured the attention of researchers over the last decade. Kinnamon et al. [26] proposed a highly sensitive portable biosensor for monitoring cortisol in human sweat using MoS2 nanosheets integrated into a flexible nanoporous electrode system. Cortisol sensing was achieved by measuring impedance changes associated with cortisol binding along the MoS2 nanosheet interface using electrochemical impedance spectroscopy. Compared with a commercial cortisol-sensing instrument, the device achieved an R^2^ correlation value of 0.998. The robustness and reliability of the device regarding commercialization were left for further study. Liu et al. [27] designed a cortisol biosensor using multilayer films containing two-dimensional tin disulfide nanoflakes, cortisol antibody, and bovine serum albumin coating on glassy carbon electrodes. Applying the designed biosensor to saliva, cortisol concentrations were determined using cyclic voltammetry. Compared with cortisol measurement using enzyme-linked immunosorbent assay, this sensor demonstrated a high R^2^ value of 0.9979, with a detectable range from 100 pM to 100 μM, a detection limit of 100 pM, and a sensitivity of 0.0103 mA/Mcm^2^. Halima et al. [28] studied a novel cortisol biosensor based on the capacitive structure of hafnium oxide, which is capable of binding with polyclonal antibodies. The interaction between cortisol and its corresponding polyclonal antibody (pAb) reflects the cortisol concentration, which can be measured by electrochemical impedance spectroscopy (EIS). This biosensor can detect cortisol in a wide range of concentrations, from 2 to 50 ng/mL, with a high selectivity. Dalirirad et al. [29] proposed a low-cost high-sensitivity aptamer-based lateral flow biosensor using the mechanism of duplex DNA dissociation to quantify the cortisol levels in human saliva. A test strip was mounted on an adhesive backing card where fluid flowed through. The detection of cortisol concentrations in human saliva ranging from 0.5 to 15 ng/mL with a limit of detection of 0.37 ng/mL has been reported.

A number of studies have applied molecularly imprinted polymers and the electrochemical technique to the detection of cortisol molecules. For example, Manickam et al. [21] used electropolymerized polypyrrole (PPy) on the surface of an electrode and evaluated the sensor’s performance with cyclic voltammetry The detection limit was 1 pM and the sensor could be rebound up to seven times before its performance was decreased by 10%. This demonstrates the rebinding ability of cortisol-imprinted polymers (CIPs), which can be a benefit in biomedical applications. Mugo et al. [30] proposed a non-enzymatic flexible biomimetic MIP-based sensor to detect cortisol in sweat using glycidyl methacrylate-co ethylene glycol dimethacrylate as a polymer. This sensor had a limit of detection of 2.0 ± 0.4 ng/mL and a detectable range of 10–66 ng/mL, with a high selectivity and reproducibility. Suda et al. [31] designed a dual-binding-site MIP-based sensor using cortisol-21-monomethacrylate and itaconic acid as the template molecule and functional monomer. The sensor was demonstrated to detect cortisol in saliva samples with a high sensitivity and selectivity and a limit of detection of 4.8 pM.

This study was focused on an MIP-based biosensor for cortisol detection. The salient aspects and/or contributions of this paper are enumerated as follows:This paper presents a cortisol biosensor with a high specificity and sensitivity, with a limit of detection by calculation at 1.035 and a performance in the wide detectable range from 0.1 pg/mL to 1000 pg/mL, which are sufficient for clinical application.The study explored the optimal polymer ingredients in order to propose an optimized polymer for use as an MIP-based cortisol biosensor.This research explored an MIP-based biosensor using a carbon electrode and gold electrode with conditions of methacrylic acid (MAA) and methyl methacrylate (MMA) monomers to optimize the conditions for the MIP-based cortisol sensor.

## 2. Materials and Methods

The overall fabrication process of the MIP-based biosensor for cortisol detection is shown in Figure 1. A pre-polymer was synthesized from a set of monomers combined with cross-linker, initiator, and solvent. After completion, the pre-polymer was mixed with graphene oxide, dropped on the sensor, and then imprinted with a cortisol template. After complete polymerization, the template of cortisol was removed from the electrode to derive the MIP-based cortisol biosensor. To detect cortisol levels, solutions containing cortisol in various concentrations were dropped on the electrode within 3 min. Detection by cyclic voltammetry using a potentiostat was then performed to measure the relative level of cortisol.

A screen-printed electrode (SPE) is an electrochemical sensor providing a simple, cost-effective, and rapid time response in real-time detection, projecting many advantages for biomolecule detection [32]. In this research, a screen-printed electrode (SPE) with two different types of working electrode, including carbon (DRP-110, Dropsens, Asturias, Spain) and gold (DRP-220BT, Dropsens, Asturias, Spain), was selected. SPEs allow for rapid on-site analysis with a high reproducibility, sensitivity, and accuracy. Specially designed to work with microvolumes, this inexpensive miniaturized solution presents a high reproducibility, making it ideal for working in decentralized assays or for developing biosensors in clinical applications. A sensitive layer was coated on the working SPE with a 4 mm diameter. A carbon electrode is a cheap metal with adequate sensing, a vigorous surface chemistry, a high potential, and a minimal background current, making it a good choice for solid electrodes. So, the carbon SPE material was chosen for research in varying conditions of polymer synthesis for the development of high-performance biosensors. However, the carbon electrode has some drawbacks in terms of exhibiting less electron transfer compared to other metals. Therefore, the gold SPE material was also chosen for this research on biosensor optimization to increase conductivity and electron charge transfer for the best detection of the target molecule.

### 2.1. Polymer Synthesis

Functional monomers are binding molecules at the binding sites of the template that were well-selected in order to increase the specificity and selectivity of the MIP. Hydrocortisone has hydroxyl groups at positions 11, 17, and 21 on its structure as negative surfaces. To synthesize the polymers used in the MIP-based cortisol biosensor, the following four functional monomers were selected: methacrylic acid (MAA), acrylamide (AAM), methyl methacrylate (MMA) positive monomer, and N-vinylpyrrolidone (NVP) hydrophobic monomer. NVP was used for maintaining the shape of the complex. The chemical structures of each monomer are provided in Figure 2, respectively. To make a homogeneous solution, cross-linker and solvent were added. To optimize the monomer ingredients for the screen-printed carbon electrode, 7 conditions to synthesize the pre-polymers used in the MIP-based cortisol biosensor were set, as shown in Table 1. Note that, as MAA and MMA are liquid functional monomers with different densities, the actual volume used in polymer synthesis can be computed as follows:(1)$$\rho =\frac{m}{V}$$
where ρ is intensity (g/cm^3^), as provided in Table 1, *m* is mass (g), and *V* is volume (µL).

In this investigation, the functional monomers were combined with 47 mg of N,N-(1,2-dihydroxyethylene) bis-acrylamide (DHEBA) as a cross-linker, which is important for controlling the morphology of the polymer matrix and stabilizing the imprinted binding site; 1.5 mg of azobisisobutyronitrile (AIBN) as an initiator, which is important for initiating free radical polymerization; and 300 µL of dimethyl sulfoxide (DMSO) as a solvent.

The process of synthesizing the polymer in condition 1, in which methacrylic acid (MAA) and N-vinylpyrrolidone (NVP) polymer are mixed in a ratio of 2:2, is as follows:(1)Drop the components for polymer synthesis into a 1.5 mL microcentrifuge tube consisting of an initiator, cross-linker, and solvent.(2)Mix the functional monomers according to condition 1 in Table 1, using 16.86 µL of MAA and 21.40 µL of NVP.(3)Polymerize these two monomers at a temperature of 60–65 °C in a water bath and allow 10 min to reach a viscous solution.(4)Cool the synthesized polymer in an ice bath to prepare for imprinting the pre-polymer.

The polymerization process in conditions 2–7 is similar to that in condition 1. The only difference is that, in step 2, the mixing gradient is modified to match the gradient defined in Table 1 and the optimal polymers of Table 1 was selected to process on gold electrode shown in Table 2.

### 2.2. Preparation of Cortisol Stock Solution

Hydrocortisone, as shown in Figure 3, was selected to represent cortisol in this research. To prepare hydrocortisone solutions of various concentrations to be tested with the MIP-based polymer, 1 mg of hydrocortisone (C_21_H_30_O_5_) (Sigma-Aldrich 386698) was dissolved in 1 mL of 4% dimethyl sulfoxide (DMSO) to make a 1 mg/mL hydrocortisone stock solution. Because the concentrations of interest in this study ranged from 0.1 pg/mL to 1000 pg/mL, the hydrocortisone stock had to be diluted using the serial dilution approach.

### 2.3. Screen-Printed Carbon Electrode Fabrication

We fabricated screen-printed electrodes coated with carbon or gold and reduced graphene oxide (GO) as an electrode modifier in the biosensing area [33]. After finishing the synthesis of the pre-polymer solution that was described in Section 2.1, 0.15 mg/mL of graphene oxide (GO) was added at a ratio of 2:3. Then, 1 µL of pre-polymer solution that was already combined with GO was dropped on the working electrode. The template stamp was prepared by adding 1 µL of hydrocortisone on top of them and polymerizing under 254 nm UV light for 3 h, followed by placing it in an incubator at 55 °C for 15 h to complete the polymerization process. Photopolymerization is a typical process in which multifunctional monomers are transformed into a tridimensional cross-linked polymeric network by chain polymerization initiated by reactive free radicals due to ultraviolet irradiation. This was followed by thermal polymerization techniques to consider a bottom-up method that referred to the self-assembly of monomer units to produce proportionately bigger and more intricate systems through interactions between building blocks using Van der Waals forces, hydrogen bonding, and electrostatic adsorption with an appropriate temperature of around 50–70 °C overnight. For the template removal step, the working area of the electrode was removed with the template of hydrocortisone on the polymer surface by soaking in a 10% acetic acid solution, stirring by a magnetic stirrer for 30 min, and rinsing with deionized water [33]. After this step was finished, the polymer on the working electrode had rigid cavities that were specific to the hydrocortisone template. Before conducting electrical measurement, the electrode was left to dry for 10 min.

### 2.4. Electrochemical Measurement

In this study, screen-printed carbon electrodes (SPEs) (Metrohm, DropSens 110, Asturias, Spain) were used as a tool for measurement. SPEs have carbon as working electrodes and counter electrodes and silver as a reference electrode. Cortisol-imprinted polymers (CIPs) and non-imprinted polymers (NIPs) were prepared on the 4 mm diameter working electrodes. Each concentration of cortisol was prepared by serial dilution in 1:1 K_4_Fe(CN)_6_:K_3_Fe(CN)_6_ (redox couple) from the cortisol stock solution (1 mg/mL) to obtain the desired range from 0.1 pg/mL to 1000 pg/mL. Then, 100 µL of a 1:1 K_4_Fe(CN)_6_ and K_3_Fe(CN)_6_ (control) mixture was dropped to perform cyclic voltammetry (the technique used to quantify the cortisol concentration). Cyclic voltammetry was conducted using Metrohm DropSens µStat 8000 potentiostats (Metrohm Siam, Bangkok, Thailand) and DropView 8400 software on a personal computer. Each concentration was measured by a cyclic voltammogram starting from −0.3 to 0.7 V with a 50 mV/s scan rate, and an on-curve measurement tool was used to obtain the peak and potential current values. The measurement started from the blank sample (no cortisol) and proceeded to various cortisol concentrations with the MIP sensor. The signals from each concentration were measured using a built-in feature in DropView 8400.

### 2.5. Preparation of Medroxyprogesterone and Drospirenone Stock Solution

To test specificity, we used medroxyprogesterone (Umeda Co., Ltd., Bangkok, Thailand) with a molecular weight (MW) of 386.52 g/mol and drospirenone (L.B.S. Laboratory, Bangkok, Thailand) with an MW of 366.50 g/mol, which were within range of the MW of hydrocortisone, which was 362.47 g/mol. They were used as negative controls for hydrocortisone because they have a similar structure to hydrocortisone, as shown in Figure 4. Medroxyprogesterone and drospirenone were diluted in ethanol and dimethyl sulfoxide to obtain the desired concentration range and imprinted on screen-printed gold electrodes. The measurement method and techniques were the same as those used for hydrocortisone detection.

## 3. Results

### 3.1. Cyclic Voltammogram in Each Condition on Carbon Electrode

The performances of the MIPs were analyzed using the differences between the final and initial currents obtained from cyclic voltammetry on the screen-printed carbon electrodes. The measurements were collected three times from three newly fabricated electrodes. All seven ratios of MAA:AAM:MMA:NVP in conditions 1 to 7 were 2:0:0:2, 0:2:0:2, 0:0:2:2, 2:2:0:2, 2:0:2:2, 2:2:0:2, and 2:2:2:2, respectively. From the cyclic voltammograms, the blank and peak current of each concentration were measured and are shown in Table 3. To compare the performances of the electrodes, the percent relative current was used, which is defined as follows:(2)$$\%Relative\;Current=\frac{\left(I-I_{0}\right)}{I_{0}}\times 100$$
where I is the current for each MIP-based sensor with various concentrations of hydrocortisone and I_0_ is the current of the sensor when the hydrocortisone concentration is 0 pg/mL or the sensor is blank. The % relative current was measured three times to obtain the average and standard deviation and plotted as linear plots, which are shown in Figure 5. The graphs demonstrate that condition 5 (2MAA:0AAM:2MMA:2NVP) was superior to the others, with the highest slope and R^2^ value in linear regression analysis. Sensitivity is defined as the slope of the fitting line.

### 3.2. Cyclic Voltammogram in Each Condition on Gold Electrode

The experimental carbon screen-printed electrode results regarding the type of polymer demonstrated that condition 5 was the best optimal ratio, with a high R^2^ in the linear regression analysis and the highest sensitivity. In comparison, condition 4 had a slightly higher R^2^ in the linear regression analysis, but a much lower sensitivity.

Condition 5 was further explored to find the optimal ratio of each monomer with the screen-printed gold electrode, and was categorized into four sub-conditions, as shown in Table 4. The average and standard deviation were computed, and then plotted as linear plots, as shown in Figure 6.

To compare the performances of the gold electrodes, they are shown in Figure 6. The graphs demonstrate that condition 5.3 (2MAA:1MMA) was superior to the others, with the highest slope and R^2^ value in the linear regression analysis. In addition, the raw data for all conditions have shown in Figure 7.

MAA is an acid monomer and MMA is a neutral monomer. Both MAA and MMA can undergo charge–charge interactions with biomolecules and bond with the amine group. Thus, they create strong bonds with cortisol, because cortisol is a polar molecule. These polymers were reproduced three times to obtain the average and standard deviation of their currents to calculate the limit of detection. Many studies on cortisol detection conducted performance tests by initially using fish blood, where the normal cortisol level ranges from 5.65 to 26.3 ng/mL, while stress can range up to 114.6 ng/mL [34]. In humans, cortisol can be found in sweat, saliva, blood, urine, and intestinal fluid. There have been studies showing that the cortisol level in sweat ranges from 1 to 142 ng/mL [35,36]. In this case, the very common range to develop cortisol biosensors is normally from around 0.1 pg/mL to 1000 pg/mL, in order to develop a high sensitivity of detection [35,36,37,38]. The limit of detection can be calculated using the linear equation of the trendline, as follows:(3)$$Limit\;of\;detection\left(LOD\right)=3.3(\frac{\sigma }{S})$$
where σ is the standard deviation of the signals and S is the slope of the trendline.

The results for the limit of detection on the gold screen-printed electrodes in condition 5.3 (2MAA:1MMA) gave approximately 1.035 pg/mL by calculation for hydrocortisone detection.

### 3.3. Specificity Test

MIPs have many outstanding advantages, especially as artificial receptors for recognition. However, this could be one of the drawbacks of the MIP technique, where the cavities created by duplicating the shape and size of target molecules can lead to non-specific binding to similar structures, shapes, and sizes of other biomolecules [39]. A specific test was performed on molecules that are similar to the hydrocortisone molecule. We tested medroxyprogesterone acetate (sex hormone) and drospirenone. These steroid hormones were selected for comparative detection on cortisol-imprinted polymers. In the specificity test, polymers with MAA and MMA at a 2:1 ratio were synthesized and imprinted on screen-printed gold electrodes, and the hormone concentration range was from 0.1 to 1000 pg/mL. Sensitivity can be computed by slope of the response graph in Figure 8. The experimental result showed that the sensitivity of the MIP-based biosensor when tested with hydrocortisone had the highest sensitivity by slope and R^2^ value when compared with the others. The test was performed to ensure the performance of the sensor and confirm that there was no concern about non-specific binding with other similar biomolecules. The result indicated that cortisol had the significantly highest R^2^ 0.9880, where medroxyprogesterone acetate and drospirenone were 0.8481 and 0.8781, respectively.

## 4. Discussion

Hydrocortisone is the name for the hormone cortisol when supplied as a medication. This research was focused on the design and fabrication of an MIP-based sensor for hydrocortisone as a biomarker of stress-related health conditions. The results demonstrated that the sensor can be used to detect hydrocortisone with a high sensitivity and specificity. Methacrylic acid (MAA), acrylamide (AAM), methyl methacrylate (MMA), and N-vinylpyrrolidone (NVP) in various ratios were optimized, and the results demonstrated that the combination of two monomers at a ratio of 2MAA:1MMA gave the best linear performance in cortisol detection, as it achieved the highest R^2^ of 0.9892 with the carbon electrode and 0.9880 with the gold electrode. The MIP sensor with a ratio of 2MAA:1MMA on the gold electrode achieved a detection limit of 1.035 pg/mL. Herein, the MIP-based cortisol hormone biosensor showed a standout performance in comparison to MIP-based cortisol sensors in prior studies. Ulygun et al. [40] used cyclic voltammetry to measure the electrical properties of carbon electrodes with biomolecule-imprinted polymers on phosphate-buffered saline (PBS) with Fe (II)/Fe (III), showing a detection range from 0.36 pg/mL to 3624 pg/mL (R^2^ = 0.9925) and a limit of detection of 3624 pg/mL. Manickam et al. [41] used electrochemical impedance spectroscopy (EIS) to measure the electrical properties of carbon electrodes with biomolecule-imprinted polymers on human saliva, showing a detection range of 0.18–23.19 ng/mL (R^2^ = 0.993) and a limit of detection of 50 pg/mL. In comparison to this study, the limit of detection through calculation and the detection range in real experiments were 1.035 pg/mL and from 0.1 to 1000 pg/mL (R^2^ = 0.9880), respectively.

When considering that the cortisol hormone concentration in real serum from humans is 5–25 ng/mL, our proposed carbon MIP-based sensor with an imprinted 2MAA:1MMA polymer had the best performance, with concentration ranging from 0.1 to 1000 pg/mL and a detection limit of 1.035 pg/mL. Other research on anticortisol antibodies (C-Abs) using cyclic voltammetry on PBS with Fe (II)/Fe (III) reported a detection range of 3–36,000 pg/mL (R^2^ = 0.997) and a limit of detection of 10 pg/mL [42]. To demonstrate the specificity of our MIP-based hydrocortisone sensor, we tested it with medroxyprogesterone acetate and drospirenone, which have similar molecular structures to that of the cortisol hormone. The sensitivity of the MIP-based hydrocortisone biosensor with a gold electrode, when applied with hydrocortisone, was 7.0364% relative current/log (pg/mL) (Figure 8). The sensitivity of the MIP-based cortisol biosensor when tested with medroxyprogesterone and drospirenone was 0.5842 and 1.0482%relative current/log (pg/mL), respectively. The limit of cortisol detection in condition 5.3 (2MAA: 1 MMA) with a gold electrode sensor showed the best limit of detection at 1.035 pg/mL. The experimental results showed that our MIP-based hydrocortisone sensor provides not only a linear response, but also a high specificity. The limit of detection of the 2MAA:1MMA sensor would be enough to detect cortisol in real serum at 5–25 ng/mL. A blinded competition test between MIP-based cortisol sensors with cortisol and cortisol-similar molecules is needed to further investigate the effect when the real solution contains a mixture of cortisol and medroxyprogesterone. This will be left for future study.

## 5. Conclusions

This study aimed to synthesize an MIP-based hydrocortisone sensor with a high selectivity and sensitivity. An electrochemical biosensor was used to demonstrate the rapid detection of hydrocortisone by recognizing this specific molecule at concentrations ranging from 0.1 to 1000 pg/mL. A gold electrode with polymers at a ratio of 2MAA:1MMA achieved the best performance, with a detection limit of 1.035 pg/mL by calculation and an R^2^ linearity of 0.9880. The synthesis process was simple, low-cost, and rapid according to the objectives of this study. This biosensor can be applied for real detection in biological samples such as urine, saliva, or sweat and could be useful in biomedical applications.

## Figures and Tables

**Figure 1 polymers-17-00545-f001:**
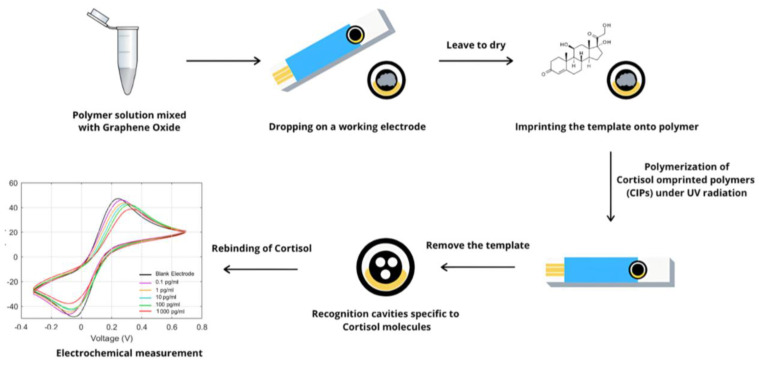
Scheme of making electrochemical biosensor for detection of cortisol. Pre-polymer solution mixed with graphene oxide is coated on working electrode and imprinted with template. After complete polymerization, template is removed, and recognition sites of cortisol are presented.

**Figure 2 polymers-17-00545-f002:**
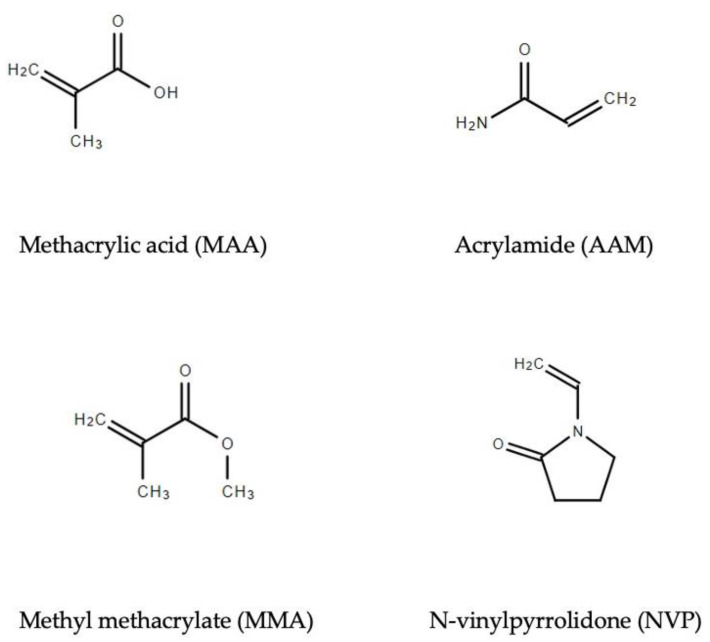
Structures of selected functional monomers.

**Figure 3 polymers-17-00545-f003:**
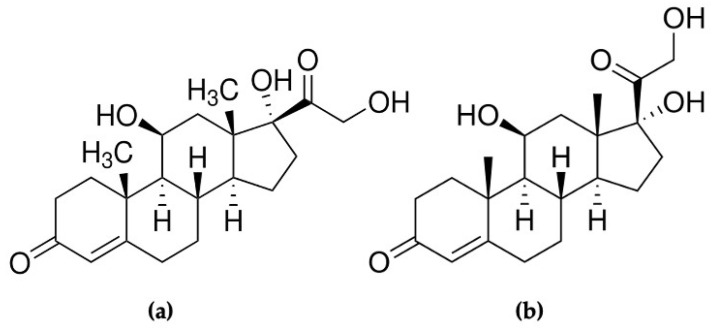
Structures of (**a**) hydrocortisone (Sigma-Aldrich 386698) compared to (**b**) cortisol.

**Figure 4 polymers-17-00545-f004:**
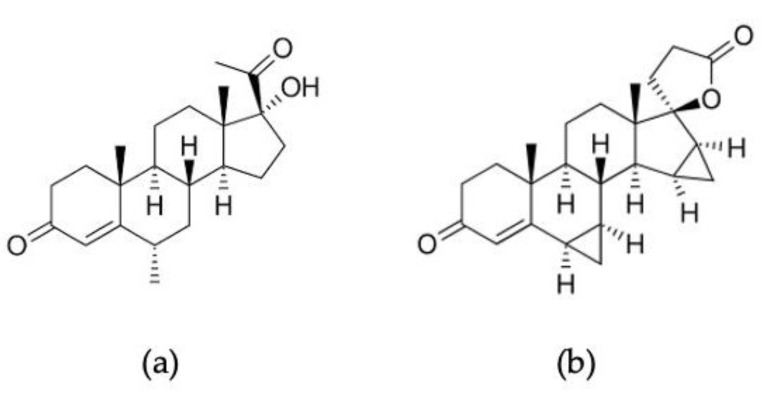
Structures of (**a**) medroxyprogesterone and (**b**) drospirenone.

**Figure 5 polymers-17-00545-f005:**
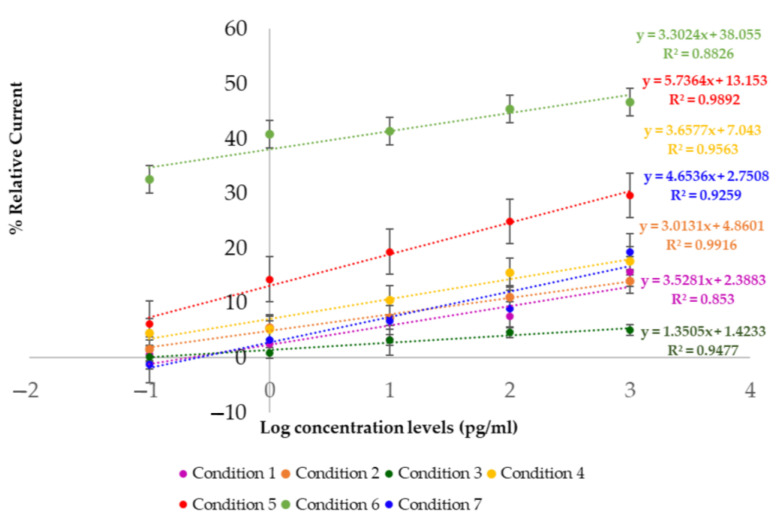
Performance of each ratio evaluated by sensitivity and R^2^ between condition 1 and condition 7, respectively. Error bars represent standard deviation.

**Figure 6 polymers-17-00545-f006:**
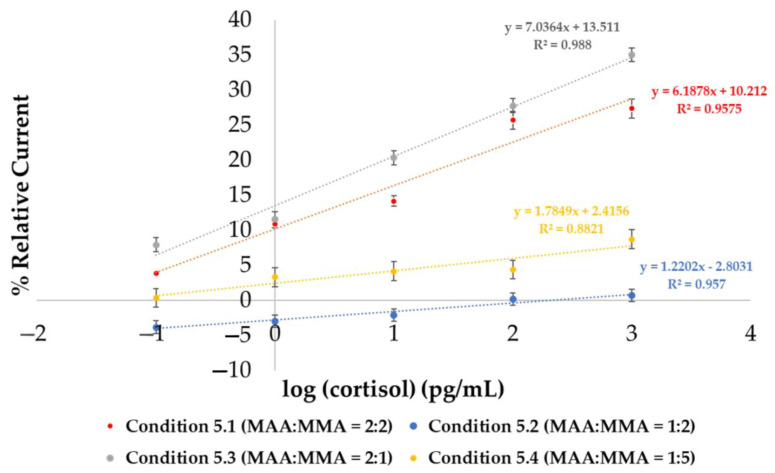
Performance of each ratio evaluated by sensitivity and R^2^ between condition 5.1 (2MAA:2MMA), condition 5.2 (1MAA:2MMA), condition 5.3 (2MAA:1MMA), and condition 5.4 (1MAA:5MMA). R^2^ values are 0.9575, 0.9570, 0.9880, and 0.8821, respectively. Error bars represent standard deviation.

**Figure 7 polymers-17-00545-f007:**
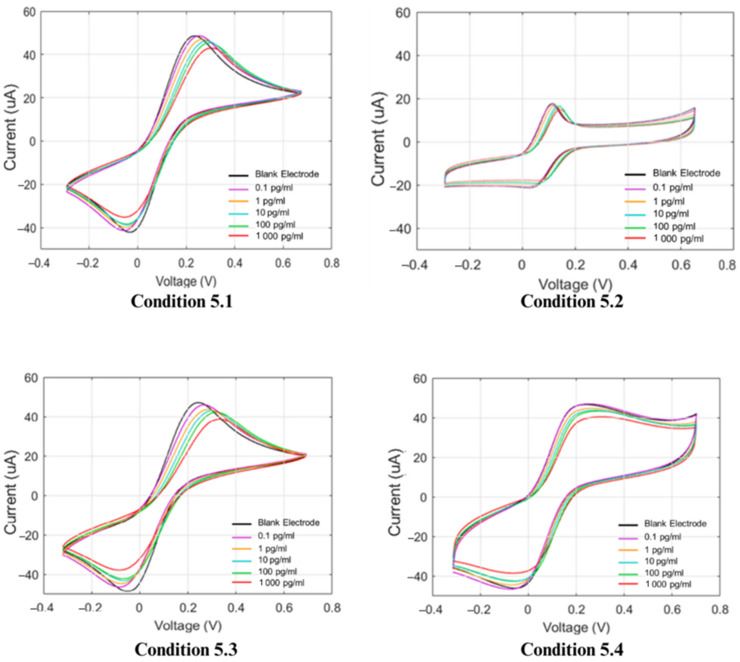
Cyclic voltammograms of conditions 5.1–5.4.

**Figure 8 polymers-17-00545-f008:**
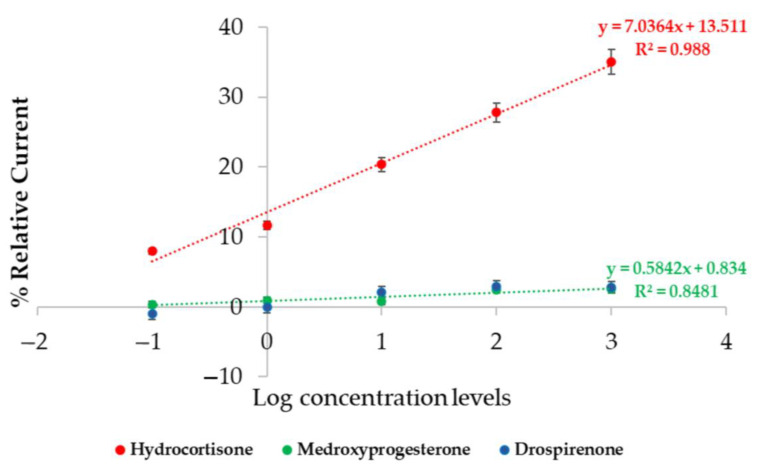
Specificity test for cortisol on carbon screen-printed gold electrode in condition (2MAA:1MMA) compared with negative control as medroxyprogesterone and drospirenone. Error bars represent standard deviation.

**Table 1 polymers-17-00545-t001:** The amount of each functional monomer in various conditions on carbon electrode.

Condition	Ratio	Methacrylic Acid (MAA) (µL)	Acrylamide (AAM) (mg)	Methyl Methacrylate (MMA) (µL)	N-Vinylpyrrolidone (NVP) (µL)
1	2:0:0:2	16.86	-	-	21.40
2	0:2:0:2	-	14.2	-	21.40
3	0:0:2:2	-	-	21.29	21.40
4	2:2:0:2	16.86	14.2	-	21.40
5	2:0:2:2	16.86	-	21.29	21.40
6	2:2:0:2	16.86	14.20	-	21.40
7	2:2:2:2	16.86	14.20	21.29	21.40

**Table 2 polymers-17-00545-t002:** The amount of each functional monomer in various conditions on screen-printed gold electrode.

Condition	Ratio	Methacrylic Acid (MAA) (µL)	Methyl Methacrylate (MMA) (µL)	N-Vinylpyrrolidone (NVP) (µL)
5.1	2:2:2	16.86	10.60	21.40
5.2	1:2:2	8.43	21.29	21.40
5.3	2:1:2	16.86	10.60	21.40
5.4	1:5:2	8.43	53	21.40

**Table 3 polymers-17-00545-t003:** The current data of conditions 1–7 on the carbon electrodes.

Condition	Concentration	Current (µA)	∆I (µA)	∆I/I_0_ × 100%
Condition 1 (MAA:AAM:MMA:NVP) (2:0:0:2)	Blank	14.634	-	-
	0.1 pg/mL	14.550	0.084	0.574006
	1 pg/mL	14.262	0.372	2.542025
	10 pg/mL	14.164	0.470	3.211699
	100 pg/mL	13.531	1.103	7.537242
	1000 pg/mL	12.334	2.300	15.716820
Condition 2 (MAA:AAM:MMA:NVP) (0:2:0:2)	Blank	61.947	-	-
	0.1 pg/mL	60.947	1.011	1.611266
	1 pg/mL	58.543	3.404	5.511672
	10 pg/mL	57.407	4.540	7.331122
	100 pg/mL	55.132	6.815	11.002321
	1000 pg/mL	53.320	8.627	13.931101
Condition 3 (MAA:AAM:MMA:NVP) (0:0:2:2)	Blank	45.634	-	-
	0.1 pg/mL	45.551	0.084	0.184073
	1 pg/mL	45.262	0.372	0.815182
	10 pg/mL	44.164	1.470	3.221282
	100 pg/mL	43.531	2.103	4.608406
	1000 pg/mL	43.334	2.301	5.040102
Condition 4 (MAA:AAM:MMA:NVP) (2:2:0:2)	Blank	48.533	-	-
	0.1 pg/mL	46.327	2.206	4.545361
	1 pg/mL	45.990	2.543	5.234397
	10 pg/mL	43.437	5.096	10.50007
	100 pg/mL	40.996	7.537	15.52964
	1000 pg/mL	39.948	8.585	17.68900
Condition 5 (MAA:AAM:MMA:NVP) (2:0:2:2)	Blank	47.569	-	-
	0.1 pg/mL	44.627	2.969	6.237919
	1 pg/mL	40.777	6.819	14.32683
	10 pg/mL	38.386	9.210	19.35037
	100 pg/mL	35.744	11.852	24.90125
	1000 pg/mL	33.492	14.104	29.63274
Condition 6 (MAA:AAM:MMA:NVP) (2:2:0:2)	Blank	15.634	-	-
	0.1 pg/mL	10.550	5.084	32.51887
	1 pg/mL	9.262	6.372	40.75732
	10 pg/mL	9.164	6.470	41.38416
	100 pg/mL	8.531	7.103	45.43303
	1000 pg/mL	8.334	7.300	46.69310
Condition 7 (MAA:AAM:MMA:NVP) (0:2:2:2)	Blank	15.634	-	-
	0.1 pg/mL	10.550	5.084	32.51887
	1 pg/mL	9.262	6.372	40.75732
	10 pg/mL	9.164	6.470	41.38416
	100 pg/mL	8.531	7.103	45.43303
	1000 pg/mL	8.334	7.300	46.69310

**Table 4 polymers-17-00545-t004:** The current data of conditions 5.1–5.4 on the gold electrodes.

Condition	Concentration	Current (µA)	∆I (µA)	∆I/I_0_ × 100%
Condition 5.1 (MAA:MMA:NVP) (2:2:2)	Blank	57.700	-	-
	0.1 pg/mL	55.483	2.217	3.842288
	1 pg/mL	51.417	6.282	10.88821
	10 pg/mL	49.511	8.189	14.19237
	100 pg/mL	47.872	14.828	25.69844
	1000 pg/mL	41.904	15.796	27.37608
Condition 5.2 (MAA:MMA:NVP) (1:2:2)	Blank	32.075	-	-
	0.1 pg/mL	33.297	−1.222	−3.80982
	1 pg/mL	33.025	−0.950	−2.96181
	10 pg/mL	32.740	−0.665	−2.07327
	100 pg/mL	32.005	0.070	0.21823
	1000 pg/mL	31.849	0.226	0.70459
Condition 5.3 (MAA:MMA:NVP) (2:1:2)	Blank	60.700	-	-
	0.1 pg/mL	55.876	4.824	7.94728
	1 pg/mL	53.650	7.050	11.61450
	10 pg/mL	48.352	12.348	20.34267
	100 pg/mL	43.831	16.869	27.79077
	1000 pg/mL	39.430	21.270	35.04119
Condition 5.4 (MAA:MMA:NVP) (1:5:2)	Blank	58.097	-	-
	0.1 pg/mL	57.881	0.216	0.37179
	1 pg/mL	56.175	1.922	3.30826
	10 pg/mL	55.679	2.418	4.16205
	100 pg/mL	55.529	2.568	4.42019
	1000 pg/mL	53.019	5.078	8.74055

## Data Availability

Data is unavailable due to privacy restrictions.

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
