# Peer review of "Electrochemical Biosensors by Means of Molecularly Imprinted Polymers (MIPs) Cortisol Recognition"

_polymers, 2025, doi:10.3390/polym17040545_

Round 1

Reviewer 1 Report

Comments and Suggestions for Authors

After a thorough review of the manuscript, I am pleased to inform you that I find the study well-conceived, scientifically rigorous, and suitable for publication after minor revision. 

1. The title would benefit from revision to better reflect the innovation and scope of the work. A revised title should make the paper more engaging, precise, and aligned with its contributions.

2. In the abstract, could you more clearly highlight the significance and novelty of this study?

3. The detection limit of 1.035 pg/mL is impressive. How does this performance translate to better outcomes in real-world clinical scenarios, such as early detection or monitoring of cortisol levels?

4. The sentence describing tests on molecules with similar structures, such as medroxyprogesterone acetate and Drospirenone, suggests a selectivity study. Could you revise this part to explicitly clarify that these tests were conducted to evaluate the selectivity of the sensor?

5. Why did you choose cyclic voltammetry (CV) for cortisol detection in this study? Techniques like differential pulse voltammetry (DPV) or square wave voltammetry (SWV) are often considered more sensitive. Was there a specific reason for not using these methods?

6. Could you please include a paragraph discussing electrochemical sensors and MIP-based electrochemical sensors in your manuscript? Specifically, it would be helpful if you could highlight the significance of these technologies and their advantages, such as their high sensitivity, selectivity, and the potential for real-time detection in various applications. 

Molecularly imprinted polymer (MIP) based electrochemical sensors and their recent advances in health applications

Electrochemical chiral sensor based on molecularly imprinted polymer for determination of (1S, 2S)-pseudoephedrine in dosage forms and biological sample

Enantioselective recognition of esomeprazole with a molecularly imprinted sol–gel-based electrochemical sensor

7. Could you kindly add a paragraph discussing carbon electrodes, gold electrodes, and screen-printed electrodes in your manuscript? It would be valuable if you could address the significance of these electrode materials and their advantages, such as their excellent conductivity, stability, and versatility in a wide range of electrochemical applications.

Advanced carbon electrode materials for molecular electrochemistry

Electrochemistry of carbon electrodes

Carbon and gold electrodes as electrochemical transducers for DNA hybridisation sensors

The interaction between DNA and three intercalating anthracyclines using electrochemical DNA nanobiosensor based on metal nanoparticles modified screen-printed electrode

8. Could you please add a few sentences discussing photopolymerization in your manuscript? It would be helpful to highlight its importance and advantages, such as its ability to precisely control polymerization through light activation, making it a highly efficient and versatile method in a variety of fields, including sensor development and material science.

Development of highly selective and sensitive molecularly imprinted polymer-based electrochemical sensors for tolvaptan assay in tablets and serum

Fabrication of an Electrochemical Sensor Based on a Molecularly Imprinted Polymer for the Highly Sensitive and Selective Determination of the Antiretroviral Drug Zidovudine in Biological Samples

Development of ultra-sensitive and selective molecularly imprinted polymer-based electrochemical sensor for L-lactate detection

9. I was wondering if any studies or work have been conducted regarding the stability of the developed sensor? It would be valuable to include any findings related to its long-term performance, reliability, and resistance to environmental factors, as this is an important aspect for its practical application.

10. I noticed that the developed sensor has not been tested on the required samples. Could you please provide some insight into why this is the case? Understanding the challenges or limitations that might have prevented this testing would be helpful for further development and evaluation of the sensor's performance.

Reviewer 2 Report

Comments and Suggestions for Authors

Summary: In this paper, the authors discuss their work regarding the development of electrochemical biosensors for cortisol detection using the molecularly imprinted polymer method. Serial combinations of polymers are tested to investigate their impacts on the sensing performance of the correspondingly fabricated biosensors. Overall, the reviewer thinks this paper could be considered for publication after the major revision.

Comments:

1. line 140. The authors should mention how long they waited before testing the cortisol solutions, which is an important metric to evaluate the performance of biosensors.

2. Figure 1 has some figure resolution issues. Please get this fixed.

3. line 155. What does the “Physical properties” mean here? This seems irrelevant to the content provided in Figure 2.

4. line 161. Why are there two symbols denoting the density?

5. Figure 2 and 5. What do those numbers mean next to chemical structures? For example, 181 in Figure 2. If they do not refer to anything, please delete them.

6. Table 1. The annotation, after the asterisk, is redundant as the same information has already been provided. Please delete.

7. Figure 4. This is unnecessary. Please remove.

8. line 234. Please remove the “deposited on the working electrode”. The next sentence describes the same thing in better detail.

9. Table 3, Figure 6, Table 4, Figure 7, and Figure 9. Please include the standard deviation in those tables and figures at the proper places. It is important to understand the device-to-device variation.

10. Figure 8. The legend is missing in all plots. Please add them.

11. Line 324 to 331. The same content is already mentioned in lines 283 to 289.

12. 3.3 Limit of detection. The reviewer thinks this subsection is unnecessary and could be merged into the last subsection.

13. Figure 9. The title, Cortisol Specificity, shall be removed here to maintain the same formatting consistency.

14. line 307. Can the authors explain why they switch from carbon electrodes to gold electrodes? There are no comparison studies performed in this work to justify this move.

Round 2

Reviewer 2 Report

Comments and Suggestions for Authors

The reviewer has no more comments and suggests accepting the manuscript in its current form.